# Hormone-Driven Temperature Optimization for Elevated Reproduction in Goldfish (*Carassius auratus*) under Laboratory Conditions

**DOI:** 10.3390/ani14182701

**Published:** 2024-09-18

**Authors:** Zeynab Taheri-Khas, Ahmad Gharzi, Somaye Vaissi, Pouria Heshmatzad, Zahra Kalhori

**Affiliations:** 1Department of Biology, Faculty of Science, Razi University, Kermanshah 6714414971, Iran; zeynab.taheri93@gmail.com (Z.T.-K.); zahrakalhori1@yahoo.com (Z.K.); 2Department of Fisheries, Faculty of Fisheries and Environmental Sciences, Gorgan University of Agricultural Sciences and Natural Resources, Gorgan 4918943464, Iran; pheshmatzad@gmail.com

**Keywords:** temperature, motility, in vitro fertilization, cryopreservation, morphology

## Abstract

**Simple Summary:**

This study investigated the use of hormones and temperature control to improve breeding success. Injecting Ovaprim significantly increased egg and sperm production. However, temperature played a critical role. A medium temperature (around 22 °C) produced the best results, with more eggs, faster egg release, and healthier sperm. Both low and high temperatures negatively impacted sperm quality and larval fish survival. Using extender E4 (15% DMSO) for cryopreservation improved fertilization rates. Overall, the study highlights the importance of precise hormone control and temperature management for successful goldfish reproduction, benefiting for fish farming.

**Abstract:**

This study investigates the efficacy of hormone-induced artificial reproduction in goldfish (*Carassius auratus*) under controlled temperatures. Ovaprim injections significantly enhanced ovulation and sperm production compared to controls. Medium temperature (22 °C) produced the highest ovulation rates, fastest ovulation timing, and optimal sperm quality (motility and morphology) compared to high (28 °C) and low (16 °C) temperature groups. The low-temperature group exhibited reduced sperm motility duration and higher rates of sperm and larvae damage. The sperm volume of the high-temperature group was higher, but their post-injection survival rates were lower. Furthermore, the lowest spawning rate and low egg quality were noted in the high temperature. Cryopreservation using extender E4 (15% DMSO) exhibited superior post-thaw sperm motility and achieved higher fertilization rates. Fertilization rates, embryo development, and larval survival were all highest at the medium temperature. Larvae hatched from fresh sperm at medium temperature exhibited faster growth and fewer deformities. These findings suggest that hormone stimulation coupled with a medium temperature regimen is critical for successful artificial reproduction in goldfish. Cryopreservation with extender E4 holds promise for sperm banking; however, further optimization is necessary to improve fertilization success with thawed sperm. Future research could explore the influence of temperature on sperm physiology and refine cryopreservation protocols to enhance fertilization rates.

## 1. Introduction

Reproduction of fish under controlled conditions necessitates specific environmental factors, including temperature, in addition to hormonal agent stimulation [1]. The synergy between these two factors enables the attainment of gamete maturation and the production of high-quality gametes and larvae in captivity [2]. It is noteworthy that optimal temperature ranges can vary among species and may be influenced by factors such as age, size, and reproductive state [3]. In the realm of aquaculture, the meticulous control and maintenance of appropriate water temperatures are not only crucial for hormonal regulation but also have a profound impact on overall growth and health. Achieving the right temperature ensures optimal growth and minimizes the risk of stress and disease [4]. Beyond its direct effects on growth and reproduction, temperature can also influence various facets of fish physiology and behavior, including metabolism, immune function, and swimming performance [5].

The goldfish (*Carassius auratus*), classified within the family Cyprinidae, serves as a significant subject for studies in reproductive biology. This is due to their relative ease of breeding and maintenance in captivity, as well as their well-defined reproductive cycle with distinct stages of gamete maturation and spawning, which can be easily observed and manipulated [6]. Goldfish is indeed a popular ornamental fish worldwide, including in Iran. Many individuals across different regions have taken up goldfish breeding as a venture, and it has proven to be a lucrative business for employers. Successful goldfish breeding requires knowledge of proper care, water conditions, and breeding techniques [7]. Natural reproduction for this species primarily occurs in the spring; however, under carefully controlled conditions, it exhibits the potential for reproduction during other seasons as well [8]. However, broodstocks often encounter reproductive failure during the off-season due to the lack of natural stimuli necessary for fish reproduction [9]. To address this issue, hormone therapy, facilitated by changes in the endocrine system specifically, the brain-pituitary-gonadal axis emerges as an efficient method for inducing sexual maturity and obtaining a significant quantity of gametes under controlled conditions [10]. Indeed, the application of hormonal stimulation in cyprinids has been shown to elevate the percentage of ovulation in females, facilitate the synchronization of spawning, and augment the number of eggs produced by a female [11,12]. In recent years, human chorionic gonadotropin hormone (HCG) and gonadotropin-releasing hormone (GnRH) have been successfully employed in the artificial reproduction of fish [13]. Ovaprim is a liquid peptide preparation consisting of a gonadotropin-releasing hormone analog and a dopamine inhibitor [14]. This hormone stimulates the release of mature oocytes or eggs in females and mature sperm in males [15]. Ovaprim is especially beneficial for species that encounter difficulties in initiating spontaneous spawning in captivity [16].

Sperm motility and morphology play pivotal roles in influencing the fertilization rate in fish [17]. In species with external fertilization, such as many fish, spermatozoa are released into the marine or freshwater environment, which not only serves as a challenging milieu but also provides signals that govern their motility [18]. Temperature emerges as a paramount environmental factor influencing sperm motility [19]. Aquatic organisms exhibit sensitivity to rapid temperature fluctuations, and deviations from the optimal range induce stress, impacting motility [20], viability [21], morphology [22], DNA [23], and ultimately, the reproductive process [24]. Sperm motility experiences an increase, to a certain extent, with rising temperatures [25]. However, exceeding a specific temperature threshold results in thermal shock, leading to the loss of sperm motility [26]. For instance, in *Acipenser baeri*, sperm motility decreases as the temperature rises from 10 to 17.5 °C [27]. In some species, like salmonids, swimming duration has demonstrated an inverse exponential relationship with water temperature [28]. Additionally, various studies have reported that lower temperatures bolster spermatozoon motility [29]. Understanding the optimal temperature conditions for fertilization holds the potential to enhance breeding programs and augment the success rate of artificial insemination [30].

Cryopreservation is a pivotal technique in the realm of aquaculture [31], offering the capacity for the long-term storage of sperm, eggs, and embryos, thereby providing a valuable resource for breeding programs [32]. The advantages of cryopreservation are manifold, encompassing the preservation of genetic diversity [33], heightened breeding program efficiency [34], and mitigated risks associated with live animal transportation [35,36]. However, cryopreservation is not exempt from its set of challenges. The freezing and thawing processes can inflict damage upon biological material, diminishing its viability and fertility [37]. To mitigate such damage, meticulous control over freezing protocols and the utilization of appropriate cryoprotectants are imperative. By prudently managing the associated risks and challenges, researchers and breeders stand to harness the benefits of cryopreservation, thereby enhancing the overall health and productivity of the fish population [38]. The success of cryopreservation is undoubtedly influenced by factors such as extender composition, cryoprotectant concentration, and freezing procedures [39]. The selection of extenders for freezing fish sperm is species-specific. Although saline and sugar solutions are commonly employed for cryopreserving several fish species [40,41], further investigations are warranted to explore sperm viability under freezing conditions, as well as the impact of diluents and cryoprotectants on sperm longevity [31].

Given the established efficacy of various hormonal interventions in optimizing fish reproduction and breeding, even outside their natural reproductive seasons, and recognizing the paramount significance of establishing precise environmental conditions, particularly temperature, this research aims to achieve the following objectives: (1) Determining the precise temperature range for optimizing hormone-induced reproductive performance. (2) Meticulously evaluating the impact of varying temperature regimes on spawning and sperm quality. (3) Focusing on improving cryopreservation protocols-based outcomes, specifically sperm motility percentage and fertilization with thawed sperm. (4) Examining the developmental progression of larvae under distinct temperature conditions. This comprehensive research approach aims to provide profound scientific insights into the field of artificial fish reproduction within controlled environments. Ultimately, it aims to advance our understanding of the intricate interplay between hormones, temperature fluctuations, and reproductive success in aquatic organisms.

## 2. Materials and Methods

### 2.1. Sample Procurement and Breeder Preparation

A total of 198 goldfish (*Carassius auratus*), averaging 2 years old, were obtained from an ornamental fish breeding center for this study. The fish consisted of 108 females and 90 males. Females averaged 35.12 g (±1.46) in weight and 13.42 cm (±1.64) in total length. Males averaged 33.26 g (±1.72) in weight and 11.71 cm (±0.99) in total length. All fish were individually housed in separate 100 L glass tanks with gentle aeration. Both males and females were individually and randomly housed in separate 100 L glass tanks equipped with gentle aeration. Fish were held in tanks containing continuously aerated (24 h) purified water treated with an aquarium filter system. Weekly, 80% of the water was renewed [42]. For the initial 7 days, all groups experienced identical lighting and temperature conditions (21–22 °C). The total test period lasted 16 days, with a consistent photoperiod of 10 h of light followed by 14 h of darkness [2]. Fish were fed a diet of carp pellets and mealworms throughout the experiment [43]. Water quality parameters (temperature, pH, and osmolality) were monitored daily to ensure they remained within optimal levels for all treatment groups [44]. To minimize potential contamination of sperm during collection, fish were fasted for two days prior to hormone injection [45].

### 2.2. Temperature Regimens and Conditions

Goldfish exhibit a wide range of temperature tolerance, surviving between 5 and 35 °C [46]. This study investigated the effects of temperature on hormone-induced reproduction. Three temperature groups were established: low (LT, 16 ± 1 °C), medium (MT, 22 ± 1 °C), and high (HT, 28 ± 1 °C) using aquarium heaters (Aquaria HT 300, IRAN AQUARIA, Tehran, Iran, corporation and research facility). Each group included 15 males and 18 females, alongside a control group without hormone injection (18 females, 15 males) at each temperature. After 24 h of temperature acclimation, all fish except the controls received hormone injections [47,48].

### 2.3. Hormone Injection Procedure and Preparation

Fish were anesthetized using a clove oil solution, a commonly used fish anesthetic, following established protocols (0.15 mL/liter) [49] The dosage of Ovaprim was determined following the manufacturer’s instructions [50]. For females, a single dose ranging from 15 to 20 µL of Ovaprim was administered, while males received a single dose of 7 to 12 µL of Ovaprim. Prior to injection, the dosage was calculated based on the weight of the fish, and normal saline was added to achieve a volume of 100 µL (approximately 0.5 mL kg^−1^ for females and 0.25 mL kg^−1^ for males) for the injection [51]. Injections were performed intramuscularly using a 1 mL syringe and a 29-gauge needle [52]. Male and female fish, which had been exposed to controlled temperature groups for 24 h, were transferred to the respective temperature groups following hormone injection.

### 2.4. Sperm Collection Procedure

Sperm were collected from each male using a micropipette. Before sperm collection, the genital area of each male was dried with paper towels to prevent contamination from water, urine, and feces [53]. The collected milt was placed into 2 mL macro-tubes, and its volume, measured in microliters (µL) [54]. To ensure the collection of all possible sperm, the abdomen of each fish was stripped. Following sperm collection, these macro-tubes were immediately placed on ice [55].

### 2.5. Sperm Motility

Fresh sperm quality was evaluated based on motility. A 10 μL aliquot of sperm was diluted 1:100 with distilled water in a micro-tube [55]. A 10 μL subsample was then placed on a slide and covered with a coverslip. Sperm motility was assessed by counting motile sperm among 100 cells under a 400× magnification Zeiss AXIO Scope A1 microscope (Carl Zeiss, Oberkochen, Germany). Five replicates were performed per sample to minimize observer bias [56,57]. An experienced observer filmed sperm motility under the microscope for later evaluation. Motility duration was measured from the initial sperm movement after adding water to the sample until at least 70% of the sperm became immotile [58]. A chronometer was used to record this duration [58,59].

### 2.6. Sperm Morphology

The Diff-Quik staining method was employed for sperm morphology evaluation [60]. In this process, for fresh sperm samples obtained in each temperature treatment were prepared at a 1:100 ratio, distributed on a thin slide, and stained after drying [61]. For the frozen sperm sample, the stored sperms were thawed after one week on steam at 37 degrees [62], and after preparing the smear, they were stained with the indicated color. The lengths of the head, tail, and total length were measured. A total head length of 200 cells per slide (one slide per male) for male sperm in each temperature treatment were measured on a light microscope, at 40× magnification. Characteristics of all samples were evaluated by the same technician [63].

### 2.7. Oocyte Collection

Fish were checked for ovulation every thirty minutes after hormone injection by gently applying abdominal pressure. When ovulation was detected, the ovulated females were dried with a towel and weighed. The sampling was performed through stripping, which involves a gentle abdominal massage in the cephalocaudal direction [64]. The oocytes were collected in a dry beaker and then weighed with an accuracy of ±0.1 g to estimate the number of oocytes produced per gram of female [65].

### 2.8. Fresh Sperm-Egg Fertilization

Each female’s ovulated eggs in each temperature treatment were manually stripped and collected individually into a dry beaker. After the oocyte stripping, a subjective (macroscopic) quality check was conducted, focusing on size uniformity, yellow color, and the absence of blood. Subsequently, sperm from each male was carefully collected using a micropipette. To enhance the quality of fertilization, sperm collected from 2–3 randomly selected males [2], were used. The number of eggs produced by each female was estimated based on egg mass weights and the number of eggs per gram of egg weight. According to the egg samples, there were approximately 900 eggs per gram. For the fertilization trial, in the MT and LT treatments, eggs extracted from each female sample were divided into several containers (each container containing 100 eggs) and mixed with the sperm of males (50 µL) from the same temperature treatment [66]. However, in the HT treatment, while a very small number of eggs were obtained, their quality was poor and unsuitable for fertilization. Therefore, no fertilization trial was conducted for the HT group. Followed by the addition of 700 mL of water to activate the sperm in the MT and LT groups. The sperm and eggs were allowed to contact each other for two minutes. Then, they were incubated in a container at the temperatures specified for MT and LT. To prevent the growth of microorganisms, 0.1 mL of methylene blue was added to the water in both MT and LT groups, and 50% of the water was replenished daily [67].

### 2.9. Fertilization Evaluation

The percentage of ovulation was calculated by dividing the number of fish producing eggs by the total number of fish in each treatment. Individual females were considered as replicates and were assigned a value of 1 for ovulation or 0 for no ovulation, respectively [68]. Relative fecundity was calculated in terms of eggs per gram of female body weight [69]. Latency time was defined as the time elapsed between female fish injection and egg stripping [70]. Furthermore, the egg survival rate up to the eyed-egg stage was used to estimate fertilization quality [71]. The percentage of eyed eggs was calculated by dividing the total number of eyed eggs by the total number of provided eggs and then multiplying the result by 100 [72].

### 2.10. Sperm Cryopreservation

Extenders were prepared using sugars such as glucose, fructose, and sucrose, along with chemicals like NaCl, KCl, and NaOH (further details can be found in Table 1). Before use, all extenders were adjusted to a pH of 7 and stored at 4 °C [73]. For each temperature group, sperm was diluted to a final concentration of 1:10 (*v*/*v*) in extenders containing the cryoprotectant dimethyl sulfoxide (DMSO) at concentrations of 15% [74]. Immediately after mixing sperm with extenders and cryoprotectants, a micropipette aliquot of 100 µL was placed into 0.25 mL cryostraws (IMV, FR). These straws were then subjected to 4 °C for 5 min, followed by exposure to liquid nitrogen (LN2) vapor at a height of 5 cm for 10 min, and finally plunged into LN2 [75,76]. Sperm specimens were thawed seven days later by immersing them in 35 °C vapor water for 5 s [77].

### 2.11. Sperm Motility Post Thminawing

To evaluate the quality of frozen sperm in each temperature treatment, and choosing the best diluent for fertilization with fresh eggs, after storage in liquid nitrogen dewar for 7 days, straws were thawed by swirling them in a 35 °C water bath for 7–10 s [77]. Then, 10 µL of it was placed on a glass slide coverslip, and its motility and duration of motility were checked with a light microscope 40×. Six replicate samples were used for each treatment [41].

### 2.12. Egg Fertilization Using Thawed Sperm

Following selection of the optimal extender based on post-thaw sperm motility, a fertilization test was conducted using thawed sperm. Similar to the fresh sperm fertilization process, eggs were collected from each temperature treatment and fertilized with sperm that had been frozen at each corresponding temperature. For thawed sperm fertilization, one or two thawed freezing straws were rapidly thawed by swirling them in a 35 °C water bath for 7–10 s. The thawed sperm was then carefully poured over the eggs [77]. The sealed end of the straw was then quickly cut with scissors, allowing the partially thawed milt-extender mixture to flow onto a Petri dish [77]. The thawed sperm was poured onto the eggs in the Petri dish. Subsequently, the dishes were incubated at the same temperature as the LT and MT groups. The percentage of eyed eggs, the total length of larvae on the seventh day, and the percentage of deformed larvae were assessed in each temperature group.

### 2.13. Larvae Evaluation following Fertilization

Seven days after fertilization, larval development was assessed in both the MT and LT groups. Larvae from fertilizations using both fresh and thawed sperm were measured for total length. Digital photographs were taken using a camera mounted on a tripod at a fixed height (20 cm). For each photo, larvae were placed in a Petri dish positioned over gridded paper to facilitate size measurement. Larval images were then analyzed using Digimizer software (version 6) to determine total length. Additionally, the number of deformed larvae in each container was counted and expressed as a percentage of the total number [78].

### 2.14. Statistical Analysis

The data’s normal distribution was confirmed using both the Kolmogorov–Smirnov and Shapiro–Wilk tests. Nonparametric data (motility rate and motility duration of sperm) were analyzed using the Kruskal–Wallis test, followed by the Mann–Whitney U comparisons test and Univariate Analysis of Variance. Parametric data (relative fecundity, percentage of eyed-eggs, sperm morphology, sperm head length, larvae total length, and abnormal larvae) were analyzed using a one-way ANOVA, followed by Tukey’s post hoc test. The data were expressed as mean ± standard deviation (SD) for all comparisons, with *p* ≤ 0.05 considered statistically significant [59].

## 3. Results

Sperm production (spermatogenesis) continued for up to 24 h after hormone injection in all temperature groups. However, the time it took to collect usable sperm (latency time) varied significantly. Males in the HT group were the fastest, producing sperm within 4–5 h. Those in the MT group required 7–8 h, and the low temperature (LT) group took the longest (10–11 h). The average volume of collected sperm was as follows: 366.66 ± 49.23 µL for HT, 354.16 ± 62.00 µL for MT, and 304.16 ± 68.94 µL for LT (*p* ≤ 0.03). Interestingly, the percentage of fish producing sperm (spermiation) was highest in the LT and MT groups (83.33% each), followed by the HT group (69.23%). Notably, no sperm were collected from males in the control groups (without hormone injection) at any temperature.

### 3.1. Sperm Motility

#### 3.1.1. Fresh Sperm

The MT group demonstrated the longest duration of sperm motility (197.5 ± 15.44 s), followed by the HT group (177.50 ± 27.01 s) and the LT group (135.00 ± 15.66 s). Regarding motility rate (%) within the first 10 s, the MT treatment had the highest percentage (97.08% ± 2.57%), followed by the HT treatment (95.83% ± 2.88%) and the LT treatment (94.58% ± 3.34%). At 30 s, HT with MT (*p* = 0.03) and HT with LT (*p* = 0.01) were significant. MT with LT were significant (*p* = 0.01) at 60 s. At 90 s, MT with LT (*p* = 0.006) and HT with LT (*p* = 0.009) were significant, and at 120 s, HT with LT (*p* = 0.00) and MT with LT (*p* = 0.00) were significant. The effects of temperature (F = 20.89, df = 2, *p* = 0.00) and time (F = 1210.33, df = 6, *p* = 0.00) were found to be significant, both separately and in combination (F = 3.80, df = 12, *p* = 0.00), on sperm motility (Figure 1).

#### 3.1.2. Thawed Sperm

Sperm motility remained consistently high throughout one week in different temperature groups when sperm were cryopreserved in E4. However, other treatments showed average post-thaw sperm motility ranging from 0 to 50%. The MT-E4 treatment exhibited the highest motility rate (%) after thawing in E4, with a delay of 10 s (91.00% ± 4.18%), followed by HT-E4 (89.00% ± 4.18%) and LT-E4 (79.00% ± 4.18%) (*p* ≤ 0.00). (Table 2). Furthermore, the MT-E4 (252.00 ± 16.43 s), LT-E4 (204.00 ± 13.41 s), and HT-E4 (180.00 ± 00 s) treatments demonstrated the longest duration of motility after thawing, respectively (*p* ≤ 0.00).

### 3.2. Sperm Morphology

#### 3.2.1. Fresh Sperm

The analysis of sperm morphology in three different temperature groups revealed that LT group had the highest percentage of sperm damage (wrinkled and detached head, bent tail, coiled tail) (33.16 ± 11.45%). Following LT, the HT group exhibited a lower significant percentage of sperm damage (15.66 ± 10.24%). MT group had the lowest percentage of sperm damage (10.41 ± 5.61%) (*p* ≤ 0.00). Additionally, measurements of sperm head length showed variations: 2.16 ± 0.25 µm in LT, 2.19 ± 0.28 µm in MT, and 2.09 ± 0.26 µm in HT (*p* ≥ 0.2) (Figure 2).

#### 3.2.2. Thawed Sperm

The analysis of sperm morphology in three different temperature groups for extender 4 revealed that LT group had the highest percentage of sperm damage (wrinkled and detached head, bent tail, coiled tail) (47.91 ± 15.12%). Following the HT group exhibited a lower significant percentage of sperm damage (27.00 ± 10.43%). MT group had the lowest percentage of sperm damage (18.50 ± 6.78%) (*p* ≤ 0.00).

### 3.3. Fertilization Quality

#### 3.3.1. Fresh Sperm

The highest ovulation rate was observed in the MT group (80%), followed by the LT group (66%), while the HT group exhibited the lowest rate (13%)) the eggs were released in small quantities on the aquarium floor. In the control group, where only temperature was applied without hormone injection, no spawning was observed. Among the temperature groups, the MT group had the shortest latency time for ovulation, with an average of 10 ± 1 h. In contrast, the LT group had a longer latency time, averaging 22 ± 1 h for ovulation. Relative fecundity, measured as eggs per gram of body weight (egg/g BW), was slightly higher in the MT group with an overall mean of 57.84 ± 19.17, compared to the LT group with 46.99 ± 16.84 (*p* ≥ 0.32) (Table 2). The first appearance of eyed-eggs occurred earlier in the MT group, approximately 48 h after fertilization. Under the LT treatment, the first-eyed egg was observed much later, at around 120 h after fertilization. The MT group showed a significantly higher hatching rate, with 65.80 ± 7.34%, when considering hatched eyed-eggs between 48–72 h after fertilization, compared to the LT group with 46.65 ± 9.98% (*p* ≤ 0.004). HT and hormone stimulation had a significant impact on fish survival rates. The HT group exhibited the lowest survival rates, with 56.41%. In contrast, the MT group showed highest survival rates, with 92.30%. The LT group had the survival rate of 94.87%.

#### 3.3.2. Thawed Sperm

Similar to fresh sperm fertilization, the first appearance of eyed-eggs occurred 48 h after fertilization in the MT group when using E4. In contrast, in the LT group, the first-eyed egg was observed much later, at around 120 h after fertilization. When considering the percentage of eyed eggs, the MT group had the highest rate, with 40.00 ± 7.93%, followed by the LT group with 13.84 ± 1.23% (*p* ≤ 0.05).

### 3.4. Larvae following Fertilization

#### 3.4.1. Fresh Sperm

In the MT group, goldfish larvae exhibited the lowest percentage of deformed larvae after hatching (8.10 ± 5.69%), whereas the LT group had the highest percentage of deformed (32.47 ± 3.09%). On the seventh day after fertilization, the total length of the larvae was the greatest in the MT group (5.14 ± 0.29 mm), followed by the LT group (3.94 ± 0.32 mm) (*p* ≤ 0.00).

#### 3.4.2. Thawed Sperm

Concerning the MT group for Extender 4, goldfish larvae exhibited the lowest percentage of deformed larvae after hatching (17.07 ± 5.14%), whereas the LT group had Extender 4, the highest percentage of deformed larvae (41.20 ± 9.43%). The longest total length of larvae seven days after fertilization with frozen sperm was observed in the MT-E4 group (4.78 ± 0.52 mm), followed by the LT-E4 group (3.54 ± 0.11 mm). According to the ANOVA results, there were significant differences in larval length after thawing frozen sperm (*p* ≤ 0.00).

## 4. Discussion

The results presented in this study demonstrated the intricate interplay between temperature and various aspects of fish reproduction, taking into account the pivotal influence of hormone injections. Specifically, we examined how temperature influences the fundamental processes of sperm and egg release, determining fertilization success, and subsequently affecting the development of fish larvae. Traditionally, goldfish breeding has been confined to specific seasons, with breeders typically maintaining a temperature range of 26–27 °C. However, attempts to induce breeding outside these customary seasons through temperature shocks often led to incomplete ovulation. In response to this challenge, we implemented hormone therapy, which proved to be a successful intervention.

In our study, we observed that maintaining an MT group proved to be the optimal condition for goldfish reproduction. The administration of Ovaprim hormone at 22 ± 1 °C resulted in an impressive 80% ovulation rate, signifying it as the most successful strategy for fertilization. This finding aligns with previous studies demonstrating the critical role of temperature in fish reproduction [48]. It is crucial to underscore the sensitivity of various fish species to rapid temperature changes, a factor that can significantly impact reproductive success. This underscores the importance of precise temperature management in aquaculture practices. Furthermore, the research conducted by Nowosad et al. [2], on *Barbus barbus* builds upon our understanding of optimal thermal conditions and the effectiveness of hormone interventions. Their results highlight that the use of CPH (common carp pituitary homogenate) and Ovaprim hormones yielded exceptional outcomes, achieving ovulation percentages ranging from 90% to 100%, with embryo survival rates at hatching reaching approximately 90%. This reinforces the potential benefits of hormone-based interventions in fish reproduction, albeit with species-specific considerations.

The observed outcomes from our experiment shed light on the significant influence of temperature on various aspects of goldfish reproduction. Under the MT condition, we noted a latency time of approximately 9 h and a hatching rate of 65.80%. In contrast, under the LT condition, the latency period extended to around 21 h, accompanied by a hatching rate of 46.65%. Interestingly, under the HT condition, the latency period was notably shorter, around 6 h. However, it is worth noting that our findings also reveal a significant limitation in the effectiveness of hormone injections at temperatures exceeding 27 °C, particularly concerning ovulation induction and survival rates in goldfish; as previously mentioned, a majority of the females failed to ovulate at this temperature, with only a few releasing a limited quantity of unfertilized eggs under abdominal pressure, which were of suboptimal quality in terms of color and size. Therefore, precise monitoring and strict control of both hormone levels and temperature are imperative when striving to optimize reproductive outcomes in aquaculture practices

This study showed that despite administering nearly identical hormone doses across male and female groups, the temperature variations yielded distinct effects on these reproductive parameters. This finding aligns with the research by Servili et al. [79] who emphasized the critical importance of the appropriate combination of inducing agent dosage and latency period in achieving optimal egg production in catfish *Clarias batrachus.* Similarly, a study by Dhara et al. [80] on *Clarias batrachus* highlighted that the highest rates of fertilization and hatching were achieved when fish were injected with carp pituitary gland extracts and kept at a temperature of 28 °C, with a latency period of 15 h. Our present study underscores that HT does not necessarily facilitate successful egg release post-hormone injection, and conversely, LT is unsuitable for fertilization in aquaculture due to prolonged latency times.

Moreover, a body of research has indicated that temperature fluctuations can either expedite or delay the growth and development of eggs and larvae [81]. For instance, Akatsu et al. [82] discovered enhanced growth in *Epinephelus tauvina* larvae at higher temperatures, while Kupren et al. [83] found that embryos of *Leuciscus leuciscus* and *L. idus* exhibited tolerance to temperatures up to 23 °C, albeit with reduced survival rates and an increased occurrence of deformities. In our current experiment, we observed that fertilized eggs with both fresh and frozen sperm became eyed-eggs in the MT group simultaneously, but this process was delayed in the LT temperature group. Additionally, eggs cultivated under MT treatment displayed normal growth compared to those under LT treatment, with a significantly lower percentage of abnormal larvae. These findings emphasize that the effects of temperature on fish reproduction are highly contingent on species, developmental stage, and prevailing environmental conditions.

Global warming presents a complex and multifaceted threat to fish reproduction, with cascading effects on marine ecosystems. Rising temperatures disrupt spawning cycles, particularly for species with limited geographic range [84]. Larval fish, especially vulnerable due to their sensitivity to environmental fluctuations, may face significant challenges under a changing climate. Studies suggest that temperature has a stronger influence on fish reproduction than elevated CO_2_ levels [85]. The ability of fish populations to adapt to these changing conditions will be critical for their long-term survival [86,87]. However, rising temperatures act as stressors, impacting fish physiology, metabolism, and behavior, potentially compromising their capacity to cope with additional environmental challenges like ocean acidification and salinity changes [88]. Understanding the combined effects of temperature and other stressors on fish stress physiology is crucial for predicting the consequences of global warming on the persistence of fish populations.

It is well established that sperm morphology is a critical factor influencing fertilization success in fish [18]. The optimal shape and size of the sperm head are essential for successful penetration through the micropyle [89]. Although research on the effects of temperature and hormone stimulation on sperm morphology in fish is relatively limited, a study by Fenkes et al. [90] on *Salmo trutta* revealed that temperature has a significant impact on head size and tail length, which in turn affect fertilization success. Considering our findings, where we observed the least damage associated with the MT treatment, which also yielded the highest success rate compared to other treatments, it can be inferred that unfavorable temperatures not only affect sperm motility but also impact the shape and size of the sperm. This reinforces the notion that temperature plays a multifaceted role in influencing reproductive processes. The reduced percentage of sperm damage in the MT group suggests that optimal temperatures may promote better sperm quality, potentially enhancing fertilization success.

Our study highlights the pivotal role of cryoprotective chemicals and extenders in safeguarding sperm viability under low-temperature conditions. Dimethyl sulfoxide (DMSO) and carbohydrates have emerged as highly effective agents for cryopreserving sperm in freshwater species [91]. Building upon this knowledge, the authors of [41] conducted an extensive investigation in common carp, underscoring the potency of CCSE 2 and DMSO in maintaining post-thaw sperm motility at an impressive 94.53%. Traditionally, the success of fish sperm cryopreservation has been evaluated based on fertilization yield, primarily focusing on motility metrics. In our experiment, Extender 4 proved to be the optimal dilution for carp sperm, displaying the highest motility percentage, longest duration, and minimal morphological damage. This selection has paved the way for successful post-thaw fertilization. Studies by [92,93] provided valuable insights into the impact of cryopreservation on sperm morphology in zebrafish and catfish. Our findings further emphasize the importance of tailoring cryopreservation solutions to the specific requirements of each species. Extender 4 exhibited fewer morphological alterations, solidifying its suitability for carp sperm preservation. Moreover, Extender 4 maintained a remarkable motility rate of over 90% even after a week, showcasing its potential for long-term preservation of highly motility and minimally damaged sperm.

## 5. Conclusions

This study explored the interplay between temperature, hormones, and goldfish reproduction. Precise temperature control was crucial, with Ovaprim at 22 °C achieving an 80% ovulation rate. Synchronization of temperature and hormones is essential to avoid breeding failure. The research also identified temperature’s influence on various reproductive aspects and the importance of sperm morphology in fertilization success. An optimal extender solution for carp sperm preservation was discovered, showing promise for long-term storage and breeding strategies. These findings offer valuable insights for optimizing aquaculture practices and potentially contribute to fish conservation efforts.

## Figures and Tables

**Figure 1 animals-14-02701-f001:**
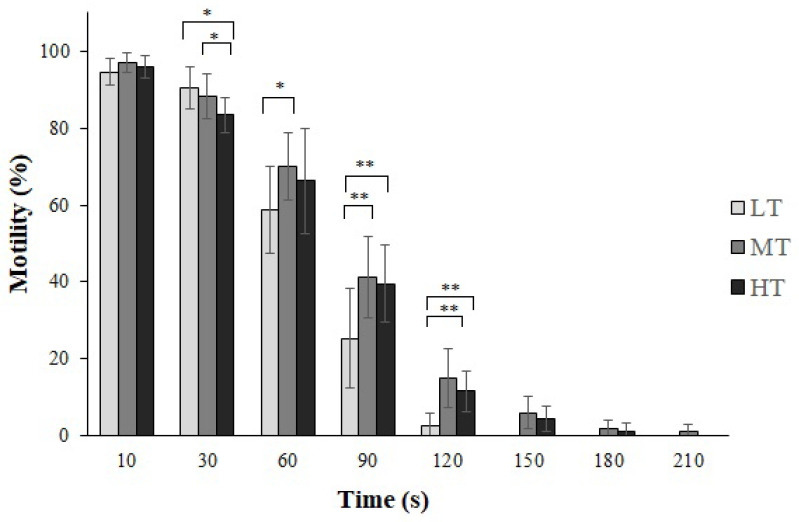
Percentage of goldfish (*Carassius auratus*) sperm motility at various temperatures. The temperatures are categorized as follows: 28 ± 1 °C (HT); 22 ± 1 °C (MT); and 16 ± 1 °C (LT). The values are mean ± SD. (*) *p*-value < 0.05; (**) *p*-value < 0.001.

**Figure 2 animals-14-02701-f002:**
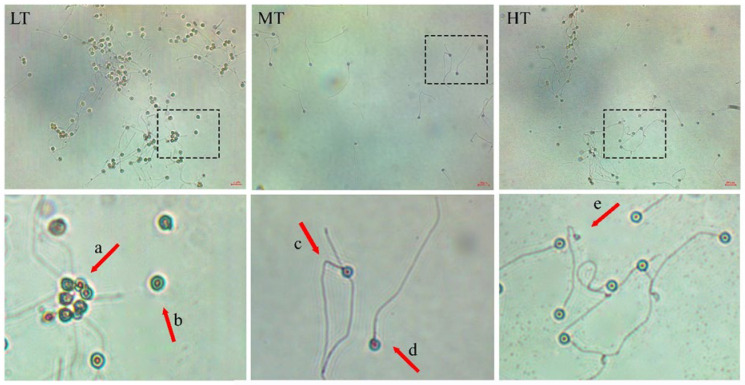
Sperm morphology of goldfish (*Carassius auratus*) in different temperature treatments. The temperatures are categorized as follows: 16 ± 1 °C (LT); 22 ± 1 °C (MT); and, 28 ± 1 °C (HT). (a) Head wrinkled; (b) detached head; (c) bent tail; (d) normal sperm; (e) coiled tail. (40× magnification).

**Table 1 animals-14-02701-t001:** The compositions of various extenders and cryoprotectants were used in the experiment.

Extenders (E)	NaCl (g)	KCl (g)	C_6_H_5_O_7_Na_3_, 2H_2_O (g)	Glucose (g)	Fructose (g)	Sucrose (g)	NaOH * (µL)	Antibiotics ** (mL)	Distilled Water (mL)	Cryoprotectant DMSO (%)
E1	0.17	0.23	-	1.06	0.9	-	12	0.5	100	10
E2	0.34	-	-	-	-	3.43	21	0.5	100	10
E3	-	0.46	-	-	1.93	-	16	0.5	100	10
E4	0.4		0.8	2.05			-	0.5	100	15

*: NaOH solution (NaOH 1 g + distilled water 100 mL). **: Antibiotic 12,000 Unit/mL penicillin.

**Table 2 animals-14-02701-t002:** The results of artificial reproduction experiments conducted with goldfish (*Carassius auratus*) at three distinct temperature conditions. The data in the table are expressed as means with corresponding standard deviations (±SD).

Temperature	16 ± 1 °C	22 ± 1 °C	28 ± 1 °C	*p*-Value
Female Weight (g)	35.15 ± 1.01	34.98 ± 2.55	35.23 ± 0.99	0.98
Ovulation (%)	66	80	13	0.001
Latency Time (h)	21–22	9–10	6–7	0.001
Relative Fecundity	46.99 ±16.84	57.84 ± 19.17	-	
Survival Rate until Eyed-Egg Stage	46.65 ± 9.98	65.80 ± 7.34	-	0.001
Percentage Deformed Larvae (%)	32.47 ± 3.09	8.10 ± 5.69	-	0.32
Total Length of Larvae (mm)	3.94 ±0.32	5.14 ± 0.29	-	0.001
Male Weight (g)	32.35 ± 1.83	34.41 ± 2.22	33.01 ± 0.56	0.37
Spermiation (%)	83.33	83.33	69.23	0.001
Sperm Motility (%)	94.58 ± 3.34	97.08 ± 2.57	95.83 ±2.88	0.001
Sperm Head Length (µm)	2.16 ± 0.25	2.19 ± 0.28	2.09 ± 0.26	0.001
Sperm Damage	33.16 ± 11.45	10.41 ± 5.61	15.66 ± 10.24	0.001
Survival Fish (%)	94.87	92.3	56.41	0.001
Post Thawing				
Sperm Motility for Extender 4 (E4) (%)	79.00 ± 4.18	91.00 ± 4.18	89.00 ± 4.18	0.001
Survival Rate until Eyed-Egg Stage	13.84 ± 1.23	40.00 ± 7.93	-	0.001
Percentage Deformed Larvae (E4) (%)	41.20 ± 9.43	17.07 ± 5.14		0.001
Total Length Larvae (mm)	3.54 ±0.11	4.78 ±0.52	-	0.001
Sperm Damage (E4)	47.91 ± 15.12	18.50 ± 6.78	27.00 ± 10.43	0.001

## Data Availability

The data presented in this study are contained within the article.

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
