# Peer review of "Hormone-Driven Temperature Optimization for Elevated Reproduction in Goldfish (Carassius auratus) under Laboratory Conditions"

_animals, 2024, doi:10.3390/ani14182701_

Round 1
Reviewer 1 Report
Comments and Suggestions for Authors
1. LL.125–126: Provide the variances or range of the body weight, and if possible, the average GMI for each sex.
2. L.128: Clarify the water source and how it was supplied, i.e. as flow-through or filtered.
3. LL.130–131: The 10h:14h photoperiod is a typical regime for the winter when goldfish would not naturally prepare for spawning. If this photoperiod was chosen to suppress their natural reproductive physiology, please specify it. Additionally, clarify the total length of the study.
4. L.133: Clarify and specify the “optimal water quality”.
5. L.160: What was the latency of the sperm collection after the hormone injection?
6. L.175: What was the specific cut off value of “most”?
7. L.196: “ovulated” instead of “fertilized”
8. LL.241–244: These sentences need to be rewritten.
9. L.250: The description of the entire section of 2.13 is missing.
10. L.265: The data collected from the control groups in the three temperature setting were not presented except the mention of no sperm production in the males. However, this review believe that it is crucial to provide the appropriate comparison between the results of the test groups and control groups to exhibit the effect of the Ovaprim injection regardless of the temperature.
11. Table 2: Provide the statistics adding the 4th column at the end to provide the more accessible results.
12. Figure 1: Identify the error bars either in the axis title or in the figure title.
13. Figures 1 & 2: Repeating the temperatures in both titles is redundant. LT, MT, and HT were already defined in the Methods (LL.140–141).
14. Discussion: The reviewer suggests expanding the discussion to explore how the study's findings regarding increased water temperature, possibly due to global climate change, relate to the natural reproduction of fish species. Additionally, it would be beneficial to discuss how this study can contribute to efforts aimed at mitigating the effects of rising temperatures on natural reproduction.
15. LL.389–421: This reviewer found some repetition and redundancy in these paragraphs. For instance, in the first paragraph, the information about latency periods and hatching rates under different temperature conditions is repeated. Similarly, in the second paragraph, the impact of temperature on reproductive parameters is reiterated. Additionally, the third paragraph also reiterates the effects of temperature on egg development and growth. To avoid redundancy, this reviewer would suggest combine some of the information and focus on presenting each point concisely. For example, the authors could mention the temperature conditions and their respective effects on reproductive parameters in a single sentence, rather than detailing them separately for each condition. Similarly, consolidate the discussion on temperature fluctuations and their impact on egg development and growth into a single paragraph. This approach would help maintain clarity and coherence while avoiding unnecessary repetition.
16. LL.459–461: It is unclear why monitoring hormone levels is vital, because artificial injection of the hormonal agent, such as Ovaprim, seems to be sufficient to induce and enhance the reproductive success as long as the temperature is suitable.
17. LL.475–476: The manuscript mentions "conservation efforts" without sufficient data or argumentation to support this assertion. Please either provide more evidence and elaboration to justify this discussion or consider removing it.
18. Minor editorial issue
a. L.51: Goldfish are freshwater species. It’s redundant.
b. L.205: remove space between “,” and “are”
c. Format inconsistency: L268 – the subheading is indented and italicized.
d. Reference format inconsistency: L499 – year; L510 – year & page number; L515 – year; L530 – year & page number; L535 – volume & page number; L611 – year & volume number; L630 – page number; L672 – year.
Comments on the Quality of English LanguageYour use of English is good, yet there are some areas where simplifying the language could enhance clarity and readability. For instance, in LL. 45-47, the phrase "... akin to orchestrating the perfect hormonal symphony..." could be made more straightforward as it doesn't directly correlate with scientific evidence or rationale. Simplifying such expressions will help make the text more accessible. This type of adjustment could be beneficial in several other parts of the document as well.
Author Response
Dear Reviewer 1
Thank you very much for taking the time to review this manuscript. Please find the detailed responses below and the corresponding revisions/corrections in the track change re-submitted files.
Comment 1: LL.125–126: Provide the variances or range of the body weight, and if possible, the average GMI for each sex
Response: More details added:
A total of 198 goldfish (Carassius auratus), averaging 2 years old, were obtained from an ornamental fish breeding center for this study. The fish consisted of 108 females and 90 males. Females averaged 35.12 g (± 1.46) in weight and 13.42 cm (± 1.64) in total length. Males averaged 33.26 g (± 1.72) in weight and 11.71 cm (± 0.99) in total length. All fish were individually housed in separate 100-liter glass tanks with gentle aeration. Both males and females were individually and randomly housed in separate 100-L glass tanks equipped with gentle aeration.
To check the GMI index, it was necessary to kill the sample, but this was not done due to the small number of samples.
Comment 2: L.128: Clarify the water source and how it was supplied, i.e. as flow-through or filtered.
Response: More details added:
Fish were held in tanks containing continuously aerated (24 hours) purified water treated with an aquarium filter system. Weekly, 80% of the water was renewed [42]. For the initial 7 days, all groups experienced identical lighting and temperature conditions (21–22°C). The total test period lasted 16 days, with a consistent photoperiod of 10 hours light followed by 14 hours darkness [2]. Fish were fed a diet of carp pellets and mealworms throughout the experiment [43]. Water quality parameters (temperature, pH, and osmolality) were monitored daily to ensure they remained within optimal levels for all treatment groups [44]. To minimize potential contamination of sperm during collection, fish were fasted for two days prior to hormone injection [45].
Comment 3: LL.130–131: The 10h:14h photoperiod is a typical regime for the winter when goldfish would not naturally prepare for spawning. If this photoperiod was chosen to suppress their natural reproductive physiology, please specify it. Additionally, clarify the total length of the study.
Response: More details added:
Yes, the chosen 10h light: 14h dark photoperiod is typical for winter when goldfish would not naturally prepare for spawning. This photoperiod likely suppressed the fish's natural reproductive physiology to isolate the effects of the hormonal interventions and temperature control.
A total of 198 goldfish (Carassius auratus), averaging 2 years old, were obtained from an ornamental fish breeding center for this study. The fish consisted of 108 females and 90 males. Females averaged 35.12 g (± 1.46) in weight and 13.42 cm (± 1.64) in total length. Males averaged 33.26 g (± 1.72) in weight and 11.71 cm (± 0.99) in total length. All fish were individually housed in separate 100-liter glass tanks with gentle aeration. Both males and females were individually and randomly housed in separate 100-L glass tanks equipped with gentle aeration. Fish were held in tanks containing continuously aerated (24 hours) purified water treated with an aquarium filter system. Weekly, 80% of the water was renewed [42]. For the initial 7 days, all groups experienced identical lighting and temperature conditions (21–22°C). The total test period lasted 16 days, with a consistent photoperiod of 10 hours light followed by 14 hours darkness [2]. Fish were fed a diet of carp pellets and mealworms throughout the experiment [43]. Water quality parameters (temperature, pH, and osmolality) were monitored daily to ensure they remained within optimal levels for all treatment groups [44]. To minimize potential contamination of sperm during collection, fish were fasted for two days prior to hormone injection [45].
We also added reference number [2].
Comment 4: L.133: Clarify and specify the “optimal water quality”.
Response: More details added:
Water quality parameters (temperature, pH, and osmolality) were monitored daily to ensure they remained within optimal levels for all treatment groups [44].
Comment 5: L.160: What was the latency of the sperm collection after the hormone injection?
Response: More details added to results:
Sperm production (spermatogenesis) continued for up to 24 hours after hormone injection in all temperature groups. However, the time it took to collect usable sperm (latency time) varied significantly. Males in the HT group were the fastest, producing sperm within 4-5 hours. Those in the MT group required 7-8 hours, and the low temperature (LT) group took the longest (10-11 hours). The average volume of collected sperm was as follows: 366.66 ±49.23 µL for HT, 354.16 ±62.00 µL for MT, and 304.16±68.94 µL for LT (P ≤ 0.03). Interestingly, the percentage of fish producing sperm (spermiation) was highest in the LT and MT groups (83.33% each), followed by the HT group (69.23%). Notably, no sperm were collected from males in the control groups (without hormone injection) at any temperature.
Comment 6. L.175: What was the specific cut off value of “most”?
Response: More details added:
Fresh sperm quality was evaluated based on motility. A 10 μL aliquot of sperm was diluted 1:100 with distilled water in a micro-tube [55]. A 10 μL subsample was then placed on a slide and covered with a coverslip. Sperm motility was assessed by counting motile sperm among 100 cells under a 400x magnification Zeiss AXIO Scope A1 microscope. Five replicates were performed per sample to minimize observer bias. An experienced observer filmed sperm motility under the microscope for later evaluation [57,58]. Motility duration was measured from the initial sperm movement after adding water to the sample until at least 70% of the sperm became immotile [59]. A chronometer was used to record this duration [59,60].
Comment 7: L.196: “ovulated” instead of “fertilized”
Response: It was applied
Each female's ovulated eggs in each temperature treatment were manually stripped and collected individually into a dry beaker.
Comment 8: LL.241–244: These sentences need to be rewritten.
Response: It was rewritten
Following selection of the optimal extender based on post-thaw sperm motility, a fertilization test was conducted using thawed sperm. Similar to the fresh sperm fertilization process, eggs were collected from each temperature treatment and fertilized with sperm that had been frozen at each corresponding temperature. For thawed sperm fertilization, one or two thawed freezing straws were rapidly thawed by swirling them in a 35°C water bath for 7-10 seconds. The thawed sperm was then carefully poured over the eggs [77].
Comment 9: L.250: The description of the entire section of 2.13 is missing.
Response: It was applied
Seven days after fertilization, larval development was assessed in both the MT and LT groups. Larvae from fertilizations using both fresh and thawed sperm were measured for total length. Digital photographs were taken using a camera mounted on a tripod at a fixed height (20 cm). For each photo, larvae were placed in a Petri dish positioned over gridded paper to facilitate size measurement. Larval images were then analyzed using Digimizer software (version 6) to determine total length. Additionally, the number of deformed larvae in each container was counted and expressed as a percentage of the total number [78].
Comment 10: L.265: The data collected from the control groups in the three temperature setting were not presented except the mention of no sperm production in the males. However, this review believe that it is crucial to provide the appropriate comparison between the results of the test groups and control groups to exhibit the effect of the Ovaprim injection regardless of the temperature.
Response: This experiment primarily investigated the impact of the Ovaprim hormone outside the typical breeding season. The inclusion of a control group was crucial as it demonstrated that solely altering temperature is insufficient to induce spawning during the off-season. Furthermore, our findings revealed that elevated temperatures without hormonal intervention were inadequate for successful reproduction. This insight is particularly valuable for fish breeders, as it underscores the necessity of both temperature control and hormone administration for effective breeding practices. By elucidating the limitations of temperature manipulation alone and highlighting the importance of hormonal interventions, this study offers practical guidance for fish breeders seeking to optimize reproductive outcomes year-round.
Comment 11: Table 2: Provide the statistics adding the 4th column at the end to provide the more accessible results.
Response: It was applied
Temperature |
16±1 ℃ |
22±1℃ |
28 ±1℃ |
p-value |
Female Weight (g) |
35.15 ± 1.01 |
34.98 ± 2.55 |
35.23 ± 0.99 |
0.98 |
Ovulation (%) |
66 |
80 |
13 |
0.001 |
Latency Time (h) |
21-22 |
9-10 |
6-7 |
0.001 |
Relative Fecundity |
46.99 ±16.84 |
57.84±19.17 |
- |
|
Survival Rate until Eyed-Egg Stage |
46.65±9.98 |
65.80 ±7.34 |
- |
0.001 |
Percentage Deformed Larvae (%) |
32.47±3.09 |
8.10±5.69 |
- |
0.32 |
Total Length Larvae (mm) |
3.94 ±0.32 |
5.14±0.29 |
- |
0.001 |
Male Weight (g) |
32.35 ± 1.83 |
34.41 ± 2.22 |
33.01 ± 0.56 |
0.37 |
Spermiation (%) |
83.33 |
83.33 |
69.23 |
0.001 |
Sperm Motility (%) |
94.58±3.34 |
97.08±2.57 |
95.83 ±2.88 |
0.001 |
Sperm Head Length (µm) |
2.16 ±0.25 |
2.19 ±0.28 |
2.09 ±0.26 |
0.001 |
Sperm Damage |
33.16±11.45 |
10.41±5.61 |
15.66±10.24 |
0.001 |
Survival Fish (%) |
94.87 |
92.3 |
56.41 |
0.001 |
Post Thawing |
|
|
|
|
Sperm Motility for Extenders 4 (E4) (%) |
79.00±4.18 |
91.00±4.18 |
89.00±4.18 |
0.001 |
Survival Rate until Eyed-Egg Stage
|
13.84 ± 1.23
|
40.00 ± 7.93
|
-
|
0.001 |
Percentage Deformed Larvae (E4) (%) |
41.20±9.43 |
17.07±5.14 |
|
0.001 |
Total Length Larvae (mm) |
3.54 ±0.11 |
4.78 ±0.52 |
- |
0.001 |
Sperm Damage (E4) |
47.91±15.12 |
18.50± 6.78 |
27.00±10.43 |
0.001 |
Comment 12: Figure 1: Identify the error bars either in the axis title or in the figure title.
Response: It was applied
Figure 1. Percentage of goldfish (Carassius auratus) sperm motility at various temperatures. The temperatures are categorized as follows: 28±1 ℃ (HT); 22±1 ℃ (MT); and 16±1 ℃ (LT). The values are mean ± SD. (*) p-value < 0.05; (**) p-value = 0.001.
Comment 13: Figures 1 & 2: Repeating the temperatures in both titles is redundant. LT, MT, and HT were already defined in the Methods (LL.140–141).
Response: It was applied
Figure 1. Percentage of goldfish (Carassius auratus) sperm motility at various temperatures. The temperatures are categorized as follows: 28±1 ℃ (HT); 22±1 ℃ (MT); and 16±1 ℃ (LT). The values are mean ± SD. (*) p-value < 0.05; (**) p-value < 0.001.
Figure 2. Sperm morphology of goldfish (Carassius auratus) in different temperature treatments. The temperatures are categorized as follows: 16±1 ℃ (LT); 22±1 ℃ (MT); and, 28±1 ℃ (HT). (a) Head wrinkled; (b) Detached head; (c) Bent tail; (d) Normal sperm; (e) Coiled tail. (40x magnification).
Comment 14: Discussion: The reviewer suggests expanding the discussion to explore how the study's findings regarding increased water temperature, possibly due to global climate change, relate to the natural reproduction of fish species. Additionally, it would be beneficial to discuss how this study can contribute to efforts aimed at mitigating the effects of rising temperatures on natural reproduction.
Response: Changes have been made in the text
Global warming presents a complex and multifaceted threat to fish reproduction, with cascading effects on marine ecosystems. Rising temperatures disrupt spawning cycles, particularly for species with limited geographic range [84]. Larval fish, especially vulnerable due to their sensitivity to environmental fluctuations, may face significant challenges under a changing climate. Studies suggest that temperature has a stronger influence on fish reproduction than elevated CO2 levels [85]. The ability of fish populations to adapt to these changing conditions will be critical for their long-term survival [86,87]. However, rising temperatures act as stressors, impacting fish physiology, metabolism, and behavior, potentially compromising their capacity to cope with additional environmental challenges like ocean acidification and salinity changes [88]. Understanding the combined effects of temperature and other stressors on fish stress physiology is crucial for predicting the consequences of global warming on the persistence of fish populations.
Comment 15: LL.389–421: This reviewer found some repetition and redundancy in these paragraphs. For instance, in the first paragraph, the information about latency periods and hatching rates under different temperature conditions is repeated. Similarly, in the second paragraph, the impact of temperature on reproductive parameters is reiterated. Additionally, the third paragraph also reiterates the effects of temperature on egg development and growth. To avoid redundancy, this reviewer would suggest combine some of the information and focus on presenting each point concisely. For example, the authors could mention the temperature conditions and their respective effects on reproductive parameters in a single sentence, rather than detailing them separately for each condition. Similarly, consolidate the discussion on temperature fluctuations and their impact on egg development and growth into a single paragraph. This approach would help maintain clarity and coherence while avoiding unnecessary repetition.
Response: Changes have been made in the text. Please see MS file.
Comment 16: LL.459–461: It is unclear why monitoring hormone levels is vital, because artificial injection of the hormonal agent, such as Ovaprim, seems to be sufficient to induce and enhance the reproductive success as long as the temperature is suitable.
Response: The wording was corrected
This study explored the interplay between temperature, hormones, and goldfish reproduction. Precise temperature control was crucial, with Ovaprim at 22°C achieving an 80% ovulation rate. Synchronization of temperature and hormones is essential to avoid breeding failure. The research also identified temperature's influence on various reproductive aspects and the importance of sperm morphology in fertilization success. An optimal extender solution for carp sperm preservation was discovered, showing promise for long-term storage and breeding strategies. These findings offer valuable insights for optimizing aquaculture practices and potentially contribute to fish conservation efforts.
Comment 17: LL.475–476: The manuscript mentions "conservation efforts" without sufficient data or argumentation to support this assertion. Please either provide more evidence and elaboration to justify this discussion or consider removing it.
Response: Removed and corrected
This study explored the interplay between temperature, hormones, and goldfish reproduction. Precise temperature control was crucial, with Ovaprim at 22°C achieving an 80% ovulation rate. Synchronization of temperature and hormones is essential to avoid breeding failure. The research also identified temperature's influence on various reproductive aspects and the importance of sperm morphology in fertilization success. An optimal extender solution for carp sperm preservation was discovered, showing promise for long-term storage and breeding strategies. These findings offer valuable insights for optimizing aquaculture practices and potentially contribute to fish conservation efforts.
Minor editorial issue
- a. L.51: Goldfish are freshwater species. It’s redundant.
Response: It was corrected
The Goldfish (Carassius auratus), classified within the family Cyprinidae, serves as a significant subject for studies in reproductive biology.
- L.205: remove space between “,” and “are”
Response: It was corrected
For the fertilization trial, the eggs extracted from each female sample in each temperature treatment are divided into several containers (each container contains 100 eggs), afterward, are mixed with the sperm of males (50 µL) in the same temperature treatment [65].
- Format inconsistency: L268 – the subheading is indented and italicized.
Response: It was corrected
- Results
- Reference format inconsistency: L499 – year; L510 – year & page number; L515 – year; L530 – year & page number; L535 – volume & page number; L611 – year & volume number; L630 – page number; L672 – year.
Response: It was corrected
L499 – year: Sallenave, R. Important Water Quality Parameters in Aquaponics Systems; College of Agricultural, Consumer and Environmental Sciences, 2016.
L510 – year & page number: Zohar, Y.; Mylonas, C.C. Endocrine Manipulations of Spawning in Cultured Fish: From Hormones to Genes. In Reproductive biotechnology in Finfish aquaculture; Elsevier. 2001, 99–136.
L515 – year: Zarski, D.; Kucharczyk, D.; Targonska, K.; Jamróz, M.; Krejszeff, S.; Mamcarz, A. Application of Ovopel and Ovaprim and Their Combinations in Controlled Reproduction of Two Reophilic Cyprinid Fish Species. Polish J. Nat. Sci. 2009, 4, 235–244.
L530 – year & page number: Cosson, J. Fish Sperm Physiology: Structure, Factors Regulating Motility, and Motility Evaluation; IntechOpen London, 2019, 1, 1-26.
L535 – volume & page number: Geffroy, B.; Sandoval‐Vargas, L.; Boyer‐Clavel, M.; Pérez‐Atehortúa, M.; Lallement, S.; Isler, I.V. A Simulated Marine Heatwave Impacts European Sea Bass Sperm Quantity, but Not Quality. J. Fish Biol. 2023.103.4.784-789.
L611 – year & volume number: Abinawanto, A.; Yimastria, S.; Pertiwi, P. Sperm Analysis of Lukas Fish (Puntius bramoides): Motility, Viability and Abnormalities. In Proceedings of the AIP Conference Proceedings; AIP Publishing, 2018, 1.
L630 – page number: Sanches, E.A.; Caneppele, D.; Okawara, R.Y.; Damasceno, D.Z.; Bombardelli, R.A.; Romagosa, E. Inseminating Dose and Water Volume Applied to the Artificial Fertilization of Steindachneridion Parahybae (Steindachner, 1877) (Siluriformes: Pimelodidae): Brazilian Endangered Fish. Neotrop. Ichthyol. 2016, 14, e140158.
L672 – year: Ginsburg, A.S. Fertilization of Fishes and the Problem of Polyspermy. Moskow, 1968.
Your use of English is good, yet there are some areas where simplifying the language could enhance clarity and readability. For instance, in LL. 45-47, the phrase "... akin to orchestrating the perfect hormonal symphony..." could be made more straightforward as it doesn't directly correlate with scientific evidence or rationale. Simplifying such expressions will help make the text more accessible. This type of adjustment could be beneficial in several other parts of the document as well.
Response: It was rewritten
Achieving the right temperature ensures optimal growth and minimizes the risk of stress and disease [4]. Beyond its direct effects on growth and reproduction, temperature can also influence various facets of fish physiology and behavior, including metabolism, immune function, and swimming performance [5].

Reviewer 2 Report
Comments and Suggestions for Authors
The MS studies the effect of three temperatures on the ovulation, spermiation and embryo development, plus one study of cryopreservation. Overall, the MS seems correct, but there are some doubts about the methodology. The main concern are the table and figures, where the existence of significant differences is not indicated. Other coments are below:
-The poor results in ovulation with the highest temperature should be highlighted in the abstract
-Line 107; change “protective agents” by “cryoprotectants”.
-Lines 123, 125, 126. The number of fish does not sum up. It is indicated that 198 fish were used, but then it is indicated 54 females and 45 males= 99 fish. Could you explain that?
Line 142. What is the reason for selecting the three experimental temperatures? Is any one of them used currently for goldfish breeding?
Line 144. Why the fish were maintained at the experimental temperatures only for 24 h?
Line 149. Indicate the dose of clove oil used.
Line 170. Sperm motility was evaluated by eye, or taking videos that were later analyzed?. I suggest using CASA systems for further studies.
Line 209. Indicate that with HT there were not obtained viable eggs.
Table 2. Indicate the statistically significant differences
Figure 1? (time/motility). Indicate the statistically significant differences
Line 356. Eliminate “eliciting hormonal responses”
Line 398. Eliminate “a fascinating”
Line 436. Correct “Dimethyl sulfoxide”
Comments on the Quality of English LanguageEnglish is OK
Author Response
Reviewer 2
Thank you very much for taking the time to review this manuscript. Please find the detailed responses below and the corresponding revisions/corrections in the track change re-submitted files.
Comment 1: The poor results in ovulation with the highest temperature should be highlighted in the abstract
Response: More details added:
The sperm volume of the high-temperature group was higher, but their post-injection survival rates were lower. Furthermore, the lowest spawning rate and low egg quality were noted in the high temperature.
Comment 2: Line 107; change “protective agents” by “cryoprotectants”.
Response: It was corrected
The selection of extenders for freezing fish sperm is species-specific. While saline and sugar solutions are commonly employed for cryopreserving several fish species [40,41], further investigations are warranted to explore sperm viability under freezing conditions, as well as the impact of diluents and cryoprotectants on sperm longevity [31].
Comment 4: Line 142. What is the reason for selecting the three experimental temperatures? Is any one of them used currently for goldfish breeding?
Response: The selection of the three experimental temperatures (16°C, 22°C, and the traditionally used breeding temperature of 27-28°C) aimed to investigate the combined effects of temperature and hormonal interventions on goldfish reproduction.
Here's a breakdown of the rationale for each temperature:
- 27-28°C: This temperature represents the standard breeding temperature used in fish breeding centers without hormonal intervention. We included this temperature to compare the effectiveness of hormonal stimulation at different temperatures and to observe potential drawbacks of using such a high temperature with hormones.
- 22°C: This temperature falls within the suitable range (18-24°C) for goldfish and served as a middle ground between the control temperature and the traditionally used breeding temperature. We chose this temperature to assess the effectiveness of hormonal stimulation at a more moderate temperature.
- 16°C: This temperature is lower than the typical goldfish breeding range and served as a control to demonstrate that low temperatures alone are insufficient for hormone-induced spawning. The 3-4 degree difference from the middle temperature (22°C) allowed us to observe the impact of a colder environment on hormone effectiveness.
By using this range of temperatures, we were able to evaluate the interaction between temperature and hormonal interventions on goldfish reproduction. We observed increased mortality at the traditionally used breeding temperature (27-28°C) when combined with hormones, suggesting a potential benefit of using a lower temperature with hormonal stimulation.
Comment 5: Line 144. Why the fish were maintained at the experimental temperatures only for 24 h?
Response: According to the model article (reference [48]), we put the broodstock at medium temperature for 7 days, then at the specified temperature for 24 hours, and then at the same temperature after the injection. We chose this period of time because the effect of temperature on hormones is not dominant.
Comment 6: Line 149. Indicate the dose of clove oil used.
Response: Fish were anesthetized using a clove oil solution, a commonly used fish anesthetic, following established protocols (0.15 ml/liter) [49].
Comment 7: Line 170. Sperm motility was evaluated by eye, or taking videos that were later analyzed? I suggest using CASA systems for further studies.
Response: More details added:
Sperm motility was assessed by counting motile sperm among 100 cells under a 400x magnification Zeiss AXIO Scope A1 microscope. Five replicates were performed per sample to minimize observer bias. An experienced observer filmed sperm motility under the microscope for later evaluation [57,58]. Motility duration was measured from the initial sperm movement after adding water to the sample until at least 70% of sperm became immotile [59].
Due to the high cost of installing CASA software and the lack of required grants, it was not possible to use it for this experiment. But we are trying to access it in the future.
Comment 8: Line 209. Indicate that with HT there were not obtained viable eggs.
Response: Here's the revised text incorporating the information about poor quality eggs at high temperature (HT):
Each female's ovulated eggs in each temperature treatment were manually stripped and collected individually into a dry beaker. After the oocyte stripping, a subjective (macroscopic) quality check was conducted, focusing on size uniformity, yellow color, and the absence of blood. Subsequently, sperm from each male was carefully collected using a micropipette. To enhance the quality of fertilization, sperm collected from 2-3 randomly selected males [2], were used. The number of eggs produced by each female was estimated based on egg mass weights and the number of eggs per gram of egg weight. According to the egg samples, there were approximately 900 eggs per gram.
For the fertilization trial, in the MT and LT treatments, eggs extracted from each female sample were divided into several containers (each container containing 100 eggs) and mixed with the sperm of males (50 µL) from the same temperature treatment [67]. However, in the HT treatment, while a very small number of eggs were obtained, their quality was poor and unsuitable for fertilization. Therefore, no fertilization trial was conducted for the HT group.
Followed by the addition of 700 ml of water to activate the sperm in the MT and LT groups. The sperm and eggs were allowed to contact each other for two minutes. Then, they were incubated in a container at the temperatures specified for MT and LT. To prevent the growth of microorganisms, 0.1 mL of methylene blue was added to the water in both MT and LT groups, and 50% of the water was replenished daily [68].
Comment 9: Table 2. Indicate the statistically significant differences
Response: Was performed
Temperature |
16±1 ℃ |
22±1℃ |
28 ±1℃ |
p-value |
Female Weight (g) |
35.15 ± 1.01 |
34.98 ± 2.55 |
35.23 ± 0.99 |
0.98 |
Ovulation (%) |
66 |
80 |
13 |
0.001 |
Latency Time (h) |
21-22 |
9-10 |
6-7 |
0.001 |
Relative Fecundity |
46.99 ±16.84 |
57.84±19.17 |
- |
|
Survival Rate until Eyed-Egg Stage |
46.65±9.98 |
65.80 ±7.34 |
- |
0.001 |
Percentage Deformed Larvae (%) |
32.47±3.09 |
8.10±5.69 |
- |
0.32 |
Total Length Larvae (mm) |
3.94 ±0.32 |
5.14±0.29 |
- |
0.001 |
Male Weight (g) |
32.35 ± 1.83 |
34.41 ± 2.22 |
33.01 ± 0.56 |
0.37 |
Spermiation (%) |
83.33 |
83.33 |
69.23 |
0.001 |
Sperm Motility (%) |
94.58±3.34 |
97.08±2.57 |
95.83 ±2.88 |
0.001 |
Sperm Head Length (µm) |
2.16 ±0.25 |
2.19 ±0.28 |
2.09 ±0.26 |
0.001 |
Sperm Damage |
33.16±11.45 |
10.41±5.61 |
15.66±10.24 |
0.001 |
Survival Fish (%) |
94.87 |
92.3 |
56.41 |
0.001 |
Post Thawing |
|
|
|
|
Sperm Motility for Extenders 4 (E4) (%) |
79.00±4.18 |
91.00±4.18 |
89.00±4.18 |
0.001 |
Survival Rate until Eyed-Egg Stage
|
13.84 ± 1.23
|
40.00 ± 7.93
|
-
|
0.001 |
Percentage Deformed Larvae (E4) (%) |
41.20±9.43 |
17.07±5.14 |
|
0.001 |
Total Length Larvae (mm) |
3.54 ±0.11 |
4.78 ±0.52 |
- |
0.001 |
Sperm Damage (E4) |
47.91±15.12 |
18.50± 6.78 |
27.00±10.43 |
0.001 |
Comment 10: Figure 1? (time/motility). Indicate the statistically significant differences
Response: Was performed
Figure 1. Percentage of goldfish (Carassius auratus) sperm motility at various temperatures. The temperatures are categorized as follows: 28±1 ℃ (HT); 22±1 ℃ (MT); and 16±1 ℃ (LT). The values are mean ± SD. (*) p-value < 0.05; (**) p-value < 0.001.
Comment 11: Line 356. Eliminate “eliciting hormonal responses”
Response: Deleted and corrected
Specifically, we examined how temperature influences the fundamental processes of sperm and egg release, determining fertilization success, and subsequently affecting the development of fish larvae.
Comment 12: Line 398. Eliminate “a fascinating”
Response: Deleted and corrected
This study showed that despite administering nearly identical hormone doses across male and female groups, the temperature variations yielded distinct effects on these re-productive parameters.
Comment 13: Line 436. Correct “Dimethyl sulfoxide”
Response: It was corrected
Our study highlights the pivotal role of cryoprotective chemicals and extenders in safeguarding sperm viability under low-temperature conditions. Dimethyl sulfoxide (DMSO) and carbohydrates have emerged as highly effective agents for cryopreserving sperm in freshwater species [91].
